# Impact of Educational Intervention on Cleaning and Disinfection of an Emergency Unit

**DOI:** 10.3390/ijerph17093313

**Published:** 2020-05-09

**Authors:** Bruna Andrade dos Santos Oliveira, Lucas de Oliveira Bernardes, Adriano Menis Ferreira, Juliana Dias Reis Pessalacia, Mara Cristina Ribeiro Furlan, Álvaro Francisco Lopes de Sousa, Denise de Andrade, Dulce Aparecida Barbosa, Luis Velez Lapão, Aires Garcia dos Santos Junior

**Affiliations:** 1Campus Três Lagoas, University of Mato Grosso do Sul, Três Lagoas 79600-080, Brazil; bruna.andrade1994@gmail.com (B.A.d.S.O.); lucasbernardes88@gmail.com (L.d.O.B.); a.amr@ig.com.br (A.M.F.); juliana@pessalacia.com.br (J.D.R.P.); maracristina.mga@gmail.com (M.C.R.F.); airesjr_@hotmail.com (A.G.d.S.J.); 2Network in Exposome Human and Infectious Diseases (NEHID), School of Nursing of Ribeirão Preto, University of São Paulo, Ribeirão Preto 14040-902, Brazil; dandrade@eerp.usp.br; 3Global Health and Tropical Medicine (GHTM), Instituto de Higiene e Medicina Tropical (IHMT), Universidade Nova de Lisboa, 1349-008 Lisbon, Portugal; luis.lapao@ihmt.unl.pt; 4Paulista School of Nursing, Federal University of São Paulo, São Paulo 04023-062, Brazil; dulce.barbosa@unifesp.br

**Keywords:** infection control, hospital cleaning service, cleaning products, feedback, disinfection, patient safety, continuing education

## Abstract

We aimed to evaluate the impact of an educational intervention on the surface cleaning and disinfection of an emergency room. This is an interventional, prospective, longitudinal, analytical and comparative study. Data collection consisted of three stages (Stage 1—baseline, Stage 2—intervention and immediate assessment, Stage 3—long term assessment). For the statistical analysis, we used a significance level of α = 0.05. The Wilcoxon and the Mann–Whitney test tests were applied. We performed 192 assessments in each stage totaling 576 evaluations. Considering the ATP method, the percentage of approval increased after the educational intervention, as the approval rate for ATP was 25% (Stage 1), immediately after the intervention it went to 100% of the approval (Stage 2), and in the long run, 75% of the areas have been fully approved. Stage 1 showed the existence of significant differences between the relative light units (RLU) scores on only two surfaces assessed: dressing cart (*p* = 0.021) and women’s toilet flush handle (*p* = 0.014); Stage 2 presented three results with significant differences for ATP: dressing cart (*p* = 0.014), women’s restroom door handle (*p* = 0.014) and women’s toilet flush handle (*p* = 0.014); in step III, there was no significant difference for the ATP method. Therefore, conclusively, the educational intervention had a positive result in the short term for ATP; however, the same rates are not observed with the colony-forming units (CFU), due to their high sensitivity and the visual inspection method since four surfaces had defects in their structure.

## 1. Introduction

The healthcare environment is an important reservoir and means of transmission of microorganisms, which may contribute to the increase of healthcare-associated infections (HAI). Prevention of HAI involves considering the impact of the environment on the transmission of epidemiologically relevant microorganisms such as noroviruses, methicillin-resistant Staphylococcus aureus, vancomycin-resistant Enterococcus spp, Acinetobacter spp. and other pathogens [1,2,3]. When it comes to healthcare facilities, this assertion becomes even more relevant, requiring that surface cleaning and disinfection (C&D) be systematically and carefully planned [4]. Surface C&D is fundamental to prevent infection and favors patient safety within health services [5,6,7].

For the effectiveness of C&D actions, to use diversified techniques with clear and objective monitoring methods is important, providing feedback to the responsible people, as well as interviews with the team to identify discrepancies in the C&D procedures performed. The literature points to the existence of several methods to assist in C&D, the most common being: fluorescent markers, measurement of Adenosine Triphosphate (ATP) and Colony-Forming Units counting (CFU) [5,8]. Each method has specific characteristics, such as visual inspection, which is a cheap method, but it has a more subjective accuracy. CFUs allow the identification of pathogens but require a laboratory and results are available after 48 h. As for ATP, they offer fast and quantitative results; however, they are more expensive and need swab and reading devices [9]. However, most of these studies were carried out in a hospital environment, and studies regarding the monitoring of surface C&D in outpatient and emergency services, with greater patient flow, lack considerably.

In this sense, improvements in the process of surface cleaning and disinfection imply qualification of the team responsible for this task. This can be achieved with training and continuing education programs that in various activities contribute to increasing and maintaining the quality of services and provide professional appreciation and motivation [5,6].

To our knowledge, only one study indicates the potential for transmission of microorganisms during activities in emergency medical services [7], although invasive procedures are often performed to maintain life in emergency units.

Based on this, this survey aims to assess the impact of an educational intervention on surface C&D of an emergency unit in Mato Grosso do Sul, Brazil.

## 2. Materials and Methods

### 2.1. Study Design, Local and Period

This is an interventional and comparative study, performed at two times: before and after cleaning. The study was conducted in an emergency unit in the city of Coxim, Mato Grosso do Sul (MS), Brazil, reference for five cities in the Northern region covering a population of 78,418 inhabitants. The facility has 65 beds, all exclusive for hospitalization of the Unified Health System (SUS) patients. Data collection occurred from September 2018 to March 2019, always in the morning shift, from 6:00 a.m. to 12:00 p.m.

### 2.2. Study Protocol

The choice of surfaces to be monitored considered their touching frequency, since surfaces that are most touched, either by patients or professionals, should receive a better C&D procedure [9,10]: medication preparation area; patients’ restroom door knob; patients’ toilet flush handle, and dressing cart, surfaces already evaluated in other published studies [7,8,11]. Only one researcher as participant observer performed the monitoring process of surfaces.

The monitoring method to assess the surfaces was visual inspection, CFU and ATP. Visual inspection of the surface was considered as “fail” when it presented: stains, scratches, splinters, rust, deterioration, dust and dirt, as the presence of one of these items may work as a microorganism reservoir [11,12].

Based on several studies available in the literature, an acceptable cutoff point (approved) was adopted for the counting of colony forming units (CFU) less than 2.5 cfu ‡/cm^2^ and for the measurement of the quantity of RLU by the ATP method, acceptable values (approved) less than 250 RLU [12,13,14,15,16].

For ATP monitoring, the reading in RLU was used with the Luminometer equipment 3M™ (Clean-Trace ATP System), Sumaré, SP, following the manufacturer’s recommendations. Contact plates or Rodac Plate^®^, containing trypticase soy agar (24 cm^2^) Plast company Labor Ind and Com EH Lab Ltd.a, Rio de Janeiro, RJ were used to evaluate colony CFUs [7,8,14]. The contact plates were pressed on the surface to be evaluated for 10 s, then they were introduced into an oven at 37 °C. The readings took place after 48 h, only by the main researcher of the study, with the aid of an electronic and digital colony counter (Logenr LS6000), allowing the counting of aerobic colonies [17].

The collections were performed twice a week, assessing the surfaces before and after the cleaning and nursing team performed the C&D, that is, four assessments before cleaning and four assessments after cleaning, totaling eight assessments per day and 16 per week for each method. The study consisted of three stages: Stage 1—baseline (4 weeks of collection), Stage 2—intervention (4 weeks of collection) and immediate assessment and Stage 3 (4 weeks of collection)—long-term assessment (two months after the educational intervention performed in Stage 2) [18,19,20,21]. There was a total of 192 evaluations for each phase, considering the three methods.

In the emergency room, the surface cleaning and disinfection was performed by eight hygiene professionals and 16 nurse technicians. They were all invited to participate in the training, but all the hygiene professionals (women) and 10 nurse technicians (two men and eight women) participated in the educational intervention. Although only 72% of the professionals participated in the educational intervention, in the second stage all professionals received feedback regarding the cleaning and disinfection process, after the collections by the researcher.

The educational intervention for individuals was based on other studies [7,18,19]. We used as educational method one training with expository dialogue, which was carried out in two periods (morning and afternoon), for 1 h and 30 min. The training consisted of slides, with images and information on the survival time of microorganisms on surfaces, measures for prevention and reduction of microorganism transmission, impact of infections related to health care and the team mobilization to adhere to the institution cleaning and disinfection protocol, with the standardization of practices. Moreover, some plaques from previous collections (Stage 1) were demonstrated with the growth of microorganisms, performed in a dynamic and participatory manner.

In Stage 1, we aimed to identify a scenario diagnosis, which was the surface C&D with the three monitoring methods: visual inspection, CFU count and ATP quantification performed by a study researcher. At this stage, the research participants were not notified about the study in order to avoid the Hawthorne effect, thus only managers were informed [7,18,21].

In Stage 2, we informed all participants of the research objective and developed an educational intervention with the data obtained in Stage 1. The intervention lasted 2 h, with the nursing and cleaning team, who are responsible for the process of C&D. During the educational intervention, the use of 87% polyester and 13% nylon composition microfiber cloths (The 3M Company) was standardized, the cloths were folded into 4 equal parts without excess product, with moderate friction for 15 s holding to dirt removal [7,21]. Both teams were instructed to use only one product: the disinfectant based on quaternary ammonium in combination with polymeric biguanide, which combines C&D in one step [14]. Immediately after the educational intervention, the same procedures of Stage 1 were performed, to verify if in the short term the educational intervention impacted the C&D process [7,18,21]. Therefore, the educational intervention was more than a training; it was also a standardization of practices, sanitizers and supplies that are used. In other words, in Stage 2, the team was aware of the objectives of the study and which surfaces were being monitored, and even received feedback during the collections [7,8,11,19].

Stage 3 occurred 60 days after the educational intervention in Stage 2. In order to assess the long-term impact, the same procedures of Stage 1 were also performed in this stage. At this stage, no feedback was provided to the team [7,8,11,19].

### 2.3. Institution’s Standard Protocol

The unit C&D routine was defined to be performed at every shift change, that is, three times a day, in the morning (7 am), in the afternoon (1 pm) and in the evening (7 pm), or when dirt was present. C&D was performed by the nursing and cleaning team. The unit had rinse-free disinfectant, indicated for the disinfection of fixed surfaces (floor); and 70% alcohol, which does not require rinsing, recommended for the disinfection of fixed surfaces (tables, devices, etc.), which have the following composition respectively: alkyl-dimethyl-benzyl-ammonium-chloride, emulsifier, foam adjuster, preservative, fragrance, dye and vehicle and hydrated ethyl alcohol—70% npn, according to the product label. However, we noted that other products were used, such as: dishwashing liquid, composition: 90% linear alkyl benzene sulfonic acid, ether, sodium sulfate, humectant, neutralizing, preservative, dye, essence and vehicle; and another disinfectant, composition: alkyl-dimethyl-benzyl-ammonium-chloride, emulsifier, foam adjuster, preservative, fragrance, dye and vehicle. The cleaning team used disinfectant and detergent and the nursing team used 70% alcohol. In the unit, the dilution of products used by the cleaning team was made by the cleaning employee.

The institution had several cloths available and with different textile compositions; however, the use among the employees was not standardized, each one used one, according to their preference. In the institution’s protocol, surfaces are mentioned in a generic way, only classifying them in critical, semicritical and noncritical areas. The institution’s cleaning protocol provided guidance on how to dilute and apply cleaning products, but it was not clear or written in a standardized manner. For this reason, through the educational intervention in our study, the quaternary ammonium-based product was standardized for cleaning the 4 surfaces evaluated.

### 2.4. Statistical Analysis

For the statistical analysis, we used a significance level of α = 0.05. Wilcoxon rank tests were applied, with the perspective of comparing the ATP quantification and microbial count results before and after the intervention in each of the assessed surfaces and stages. To compare the variation of microbial count and ATP quantification, the Mann–Whitney test was applied in each of the surfaces and stages evaluated. The quantitative approach aims to assess the microbial count and ATP quantification data by comparing the study stages. For this approach, quantitative data on total aerobic microbial count (CFU/cm^2^) and ATP were compared, and the variation of these data was calculated by the following expression:Variation % (ATP or CFU) = after − before/before × 100

Positive variations indicate the values collected after the intervention were higher than the values collected before the intervention. Negative variations indicate the opposite, that is, the values collected after the intervention were lower than the values collected before the intervention. Thus, positive variations indicate increase in relative light units (RLU) or CFU and negative variations indicate decrease in RLU or CFU.

The study complied with national and international standards for research with human subjects.

## 3. Results

We performed 192 assessments in each stage, considering the three monitoring methods, with 576 evaluations in total. Table 1 shows results of the quantitative data assessed by comparing the pre and postintervention situations of the four surfaces evaluated in the study. Additionally, the table shows results of the variation of quantitative variables in order to compare the methods employed (Table 1).

Table 2 shows results of the proportions found in each of the surfaces evaluated according to visual inspection. Notably, the proportions described refer to the surfaces that passed the visual test before and after cleaning, following the cutoff values for the ATP count less than 250 RLU and for the CFU count less than 2.5 CFU/cm^2^.

The results show that in Stage 1 there were no differences in the proportions of surfaces approved by the visual test, since the *p* values found for Fisher’s exact test were higher than 0.05 (*p* > 0.05). For Stage 2, there were no surfaces approved by visual analysis before and after cleaning, except for the medication preparation area (*p* = 0.007), showing that after cleaning, occurrence of approval increased significantly (from 25.0% to 100% of surfaces). Similar result was observed for the same surface in Stage 3, showing that after cleaning, the medication preparation area presented a significant increase in the approval occurrence (from 25.0% to 87.5% of the total areas evaluated). For Stage III there were no approved surfaces on visual inspection except for the medication preparation area (Table 2).

Graphs were prepared to observe the behavior of each ATP (RLU) and aerobic bacterial count (CFU) values per assessed surface and per stage. In this case, values below 250 RLU and 2.5 CFU/cm^2^ were considered as indicative of surface approval. Figure 1 shows the individual values’ graph for the ATP rates of the four surfaces in the three stages after intervention.

Microbial quantification (CFU/cm^2^) was also evaluated according to the cutoff point of 2.5 CFU/cm^2^ (Figure 2).

The analysis of results in Figure 2 allows us to suggest that the microbial count methodology is more demanding in terms of approving the evaluated surfaces. Analyzing the results obtained, only two surfaces presented 25% approval, namely: the women’s restroom door handle (in Stage 2) and the toilet flush handle (in Stage 3). In Stage 1, only the toilet flush handle presented 12.5% approval, compared with the other surfaces that presented total disapproval regarding the microbial counting method.

Thus, stylistically, Stage 2 (immediately after the educational intervention) had the best results, since it had a higher number of *p*-values below the significance level adopted for the test. Accordingly, it had more events of significant differences. Analyzing Figure 1, Stage 1 had the worst results; for Figure 2, Stage 3 had the worst results. The results of Stage 3 were worse than Stage 2, showing that in the long term (after 2 months of educational intervention) the team did not maintain adherence to the educational intervention carried out.

In general, the analysis of surfaces in relation to microbial counting is more demanding and judicious when compared with visual analysis and ATP quantification by bioluminescence, since the failure rate of surfaces using the bacterial counting method is significantly higher than the failure rates of the other two methodologies addressed.

## 4. Discussion

The results show that conducting a training program with the cleaning staff can intervene in the C&D process in an emergency service if monitoring of cleaning practices is carried out by providing feedback to the staff.

However, noticeably, the surface cleaning and disinfection process assessment can use several methods considering their advantages and disadvantages so that it is not possible to define that the success of the C&D process was the result of applying a certain method. It is interesting, however, to verify their complementation, if after the interventions the medians remained or increased and if the number of surfaces increased or decreased with statistically significant differences, allowing one to assess with objective mechanisms whether the surfaces are within the acceptable standards of cleanliness according to the cutoff points proposed in the literature and used in several studies [7,18,19].

The choice of an educational intervention program is fundamental to improving the quality of the surface C&D procedure, since the lack of compliance to C&D practices by the team and the lack of monitoring and supervision with feedback to the team directly impact the quality [21]. In this study, we observed that the institution had a professional responsibility to assess the C&D procedure, however, the educational actions were carried out occasionally and not in a continuous process.

In this sense, the definition of protocols to be standardized with the team is as important as the periodicity of educational activities, from the perspective of determining that they should be cleaned, as well as the frequency, order, decision of which products to use, the product concentration and the correct contact time for effectiveness [22]. During the educational intervention in Stage 2, the C&D protocol was updated, and we found that the training provided a moment of exchange of ideas and knowledge on the role of surface C&D in the process of HAI transmission.

In a study carried out in a health clinic to monitor the quality of the C&D procedure, it was found that the C&D effectiveness of the surfaces tested was not satisfactory [18]. In this context, some aspects are related to the negative impacts of interventions, such as those that do not adhere to the protocol; inadequacy of the procedures performed; use of contaminated materials, utensils or equipment; and lack of feedback from the managers to the team [21].

All methods have advantages and disadvantages in relation to their use, the costing of swabs and the device being the main disadvantages; however, this method allows the immediate feedback to the team [22]. It is worth highlighting the need for action planning considering the reality of the services and their characteristics.

In Stage 1 only one surface presented 100% approval (dressing cart), while in Stage 2 all surfaces were approved, and in Stage 3 only the women’s toilet flush handle showed 62.5% approval, with the other surfaces with 100% approval. In a survey conducted in an outpatient clinic [21], it was found that the rate of noncompliance in C&D considering the ATP method was 37.5% in Stage 1. After the intervention the success rate increased by 43.96% and in Stage 2 it increased by 70.6%. Two months after the intervention it increased by 76.52%. That is, it demonstrates improvement in the quality of the cleaning procedure after the educational intervention and in the long term. However, this did not happen in our study, showing that constant educational intervention is necessary.

As for CFUs, it was found that the approval rate in Stage 1 was only 12.5% for the toilet flush handle surface; in Stage 3 that same surface had a 25% approval. Most bacterial populations, regardless of context (environmental, food or medical), tend to form clusters of diverse species constituting a community of microorganisms, i.e., plaques [23,24]. Thus, we cannot overlook the fact that the microbial load on surfaces can be reduced by routine cleaning of the environment, mitigating the contamination [25,26].

Another aspect to consider is the choice of the C&D product, as the success of C&D on abiotic surfaces depends not only on the disinfectant chosen but also on the concentration of the active principle, time of action, mode of use, on which microorganisms it acts and the nature of the material to be disinfected [25]. In the baseline stage, it was observed that each professional used a product, so the protocol and educational intervention standardized the use of only one product with active principle of quaternary ammonium. The selection of cleaning products should include a review of current cleaning agents and disinfectants used in the institution [22].

Thus, apparently clean surfaces when observed with visual inspection may be disapproved by other assessment methods [11]. Health service environments deserve wide attention, as they act as probable reservoirs of microorganisms [25]. In this study, it was found that, in Stage 1, no statistical difference occurred in visual inspection, but in Stage 2 and Stage 3, it did. It is noteworthy that the dressing cart, restroom door handle and toilet flush handle surfaces were defective in their structure and were considered inadequate in relation to C&D.

Interventions immediately improved the effectiveness of cleaning; however, as the results of another study show [7], evidence shows that the positive effect of interventions is not sustained over time; the reasons for this reduction being lack of material resources, the fact that cleaning is a physically demanding activity and insufficient number of professionals. This thought goes against our study, because the same surface was often not approved, due to structural defects, which were not exchanged by the institution and had no provision for this change, thus the professionals cleaning the surfaces had to remove the excesses, but it frequently did not happen, which favored inadequate C&D.

Therefore, it is noted that education/training should be permanent for teams, with investment of institutions related to surface cleaning and disinfection, so that teams can observe and reflect on the effects of their actions when performed correctly and incorrectly, with their possible benefits and consequences. Thus, they will become aware of the importance of these actions being carried out properly.

This study has limitations, for example, the fact that it was performed in only one institution for a limited period hinders the generalizability of studies. In addition, each method specificity was also a limitation, as ATP indicates only the presence of organic matter but not necessarily microorganisms, having high sensitivity and low specificity. Some demographic variables that could not be analyzed, such as years of experience, may have influenced the results. Finally, we highlight the budgetary limitations that allowed us to analyze only four surfaces, although they are those with great support in the literature.

## 5. Conclusions

We found that commitment and clarification, such as the use of feedback to the team, is paramount in continuing education, being highly important and effective to the work and function properly developed by the staff responsible for surface C&D. Note that in the early stages they had no knowledge and the failure rates of the surfaces were higher, but immediately after educative action for the teams, the number of approved surfaces increased. In addition, the approval rate in Stage 3 for the ATP method was 100% approval.

Both findings suggest the need to maintain a constant educational intervention, as this result did not maintain in the long term. Notably, four surfaces showed high failure rates in the visual inspection method, because of the defects in the physical structure. The CFU result did not maintain in the long term either, as this is considered a more specific method. It is important to highlight the lack of standardization of practices performed by the team before the intervention, including the use of products of active principles and various purposes to perform C&D.

## Figures and Tables

**Figure 1 ijerph-17-03313-f001:**
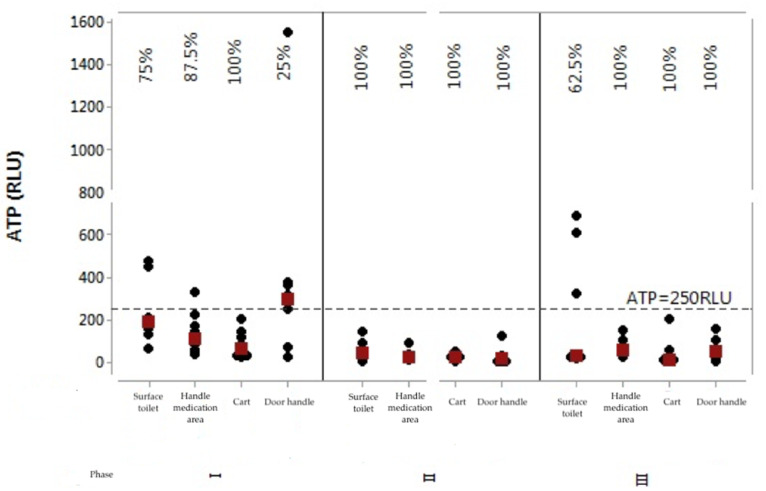
ATP values in relative light units (RLU) for surfaces in the three stages assessed. Coxim, MS, Brazil, 2018/2019. **Note:** Percentages related to approval rates. Black dots indicate individual ATP values and red dots indicate data medians.

**Figure 2 ijerph-17-03313-f002:**
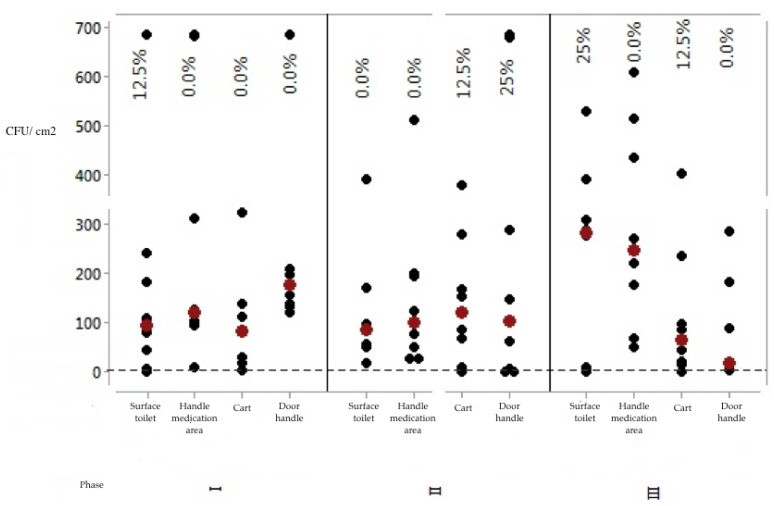
Microbial count values for surfaces in the three phases assessed. Coxim, MS, Brazil, 2018/2019. Note: Percentages related to approval rates. Black dots indicate individual CFU/cm^2^ values and red dots indicate data medians.

**Table 1 ijerph-17-03313-t001:** Median results (minimum, maximum) for Stages 1, 2 and 3 of the samples obtained from the surfaces assessed in the study. Coxim, MS, Brazil, 2018/2019.

Stage 1 (Without Intervention)									
Analysis Method	Cleaning	Medication Preparation area	*p*-Value	Dressing Cart	*p*-Value	Women’s Restroom Door Handle	*p*-Value	Women’s Toilet Flush Handle	*p*-Value
ATP (RLU) 1	Before	117 (33;343)	1.000	195 (35; 1680)	**0.021**	810.5 (505; 1123)	0.107	472 (293; 1846)	**0.014**
	After	113.5 (41;331)		62.5 (26; 205)		299 (27; 1552)		188.5 (62;477)	
Bacteria (CFU/cm^2^) 1	Before	129.5 (10;180)	0.529	315 (35; 686)	0.080	137 (50;247)	0.107	268.5 (85; 686)	0.076
	After	121 (8;686)		82.5 (3;324)		176 (120; 686)		95.5 (0;686)	
Variation Analysis 2	RLU	1.1 (−60.7; 152.8)	0.636	−50.5 (−98.1; 11.4)	0.318	−61.2 (−97.2; 99.5)	**0.010**	−73.3 (−88.6; −5.8)	0.874
	CFU	68 (−84, 1040)		−75 (−97;775)		48.5 (−51.4; 458)		−45.9 (−100; 83.3)	
**Stage 2 (After Intervention—Short Term Assessment)**									
ATP (RLU) 1	Before	63 (13;246)	0.107	159 (29;400)	**0.014**	269 (88;397)	**0.014**	535 (286;868)	**0.014**
	After	24.5 (10;91)		23.5 (8;50)		15.5 (5;123)		43.5 (7; 145)	
Bacteria (CFU/cm^2^) 1	Before	253 (65;562)	0.294	418.5 (5;686)	0.059	257.5 (45; 686)	0.726	432 (207;686)	**0.014**
	After	100.5 (26; 512)		120 (0;380)		105 (0;686)		84.5 (17;393)	
Variation Analysis 2	RLU	−61.3 (−94.4; 42.1)	0.874	−83.3 (−93.9; −25.4)	0.713	−94.7 (−98.2; −20.6)	0.636	−90.5 (−99.2; −79.1)	0.103
	CFU	−42.7 (−95.4; 687)		−79.5 (−100; 19.1)		−66 (−100,1411)		−84 (−91.8; −23.4)	
**Stage 3 (After Intervention—Long Term Assessment)**									
ATP (RLU) 1	Before	40 (20;1111)	0.624	48.5 (17; 405)	0.183	162 (51; 953)	0.080	31 (20;686)	0.363
	After	56.5 (27;149)		14.5 (7;201)		51 (5;155)		31 (20;686)	
Bacteria (CFU/cm^2^) 1	Before	265.5 (24; 686)	1.000	158.5 (73; 605)	0.107	220 (42;681)	**0.014**	594.5 (297; 684)	**0.014**
	After	246 (49;610)		64.5 (1;403)		16.5 (4;286)		283 (0;530)	
Variation Analysis 2	RLU	−5.1 (−93;645)	0.874	−73.3 (−97,415.4)	0.874	−87.6 (−99.1, 109)	0.713	−87.8 (−99; 183)	0.792
	CFU	−16 (−85; 2050)		−71.4 (−99.1; 171)		−88 (−98.4; −10.3)		−57 (−100; −2.4)	

Note: CFU: colony-forming units; ATP: adenosine triphosphate; RLU: relative light unit. 1 for Wilcoxon rank test at *p* < 0.05. 2 *p*-value for the Mann–Whitney test at *p* < 0.05. Values in bold show significant differences at *p* < 0.05.

**Table 2 ijerph-17-03313-t002:** Proportion of surfaces with approved visual result before and after intervention of hospital surfaces. Coxim, MS, Brazil, 2018/2019.

Visual Inspection		Intervention		*p* Value *
		Before	After	
Stage 1 (*n* = 8)	Medication preparation area	3 (37.5%)	2 (25.0%)	1.000
	Dressing cart	0 (0.0%)	0 (0.0%)	1.000
	Women’s Restroom Door Handle	0 (0.0%)	0 (0.0%)	1.000
	Women’s toilet flush handle	0 (0.0%)	0 (0.0%)	1.000
Stage 2 (*n* = 8)	Medication preparation area	2 (25.0%)	8 (100%)	**0.007**
	Dressing cart	0 (0.0%)	0 (0.0%)	1.000
	Women’s Restroom Door Handle	0 (0.0%)	0 (0.0%)	1.000
	Women’s toilet flush handle	0 (0.0%)	0 (0.0%)	1.000
Stage 3 (*n* = 8)	Medication preparation area	2 (25.0%)	7 (87.5%)	**0.041**
	Dressing cart	0 (0.0%)	0 (0.0%)	1.000
	Women’s Restroom Door Handle	0 (0.0%)	0 (0.0%)	1.000
	Women’s toilet flush handle	0 (0.0%)	0 (0.0%)	1.000

* value referring to Fisher ′s exact test for two proportions at *p <* 0.05.

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
