# Peer review of "Impact of Educational Intervention on Cleaning and Disinfection of an Emergency Unit"

_ijerph, 2020, doi:10.3390/ijerph17093313_

Round 1

Reviewer 1 Report

The article presented a very important problem related with effectiveness of cleaning and disinfections in emergency units. Studies were good planed and described. I have some comments, especially to the analysis of the results.

Abstract

Line23: According to statistically terminology, “p” means a probability value which appears like the result of a specific statistical test. The significance level is defined as α=0.05

Line 28: In my opinion, after “assessed” should be semi-colon

Introduction

Line 37: „Healthcare” should be written without space.

Line 40:  There should be used plural form for norovirus.

Line 41: Enterococci is not professional term, please write Enterococcus spp.

Line 41: “spp.” should be written without italics

Line 49: CFU is not a method. If authors want write about method, it should be “CFU counting”

Line 60: Authors write “Few studies…”, but there is only one reference to this sentence.

Material and Methods

In my opinion Chart No 1 in not necessary to understand number of evaluations at each stage of the study.

Line 148: p value means probability value

Line 157: I think that authors missed the parenthesis for: (after-before)

Lines 123-127: This part should be transferred to previously part of this section (after 82 line). There must be added part about interpretation criteria for CFU counting and ATP measuring.

Results

I am worried about the statistical analysis results. For example, in the table No 1 is presented p-value 0.014 for ATP measurement for women’s toilet flush handle: the median value is 472 RLU, but the distance between minimum and maximum values is high (293 – 1846). It is related probably with acceptance of outlier values. The outlier values are presented also on the figures, for example: on the Figure No 1 is marked value 1552 for door handle in first stage – in my opinion, this value should be removed from the analysis. Please check all analysis results and explain, why did you accept outliers.

I do not understand why authors decided that 250 RLU and 2.5 CFU/cm2 are cut-off values for surface approval. Please, add the reference for this criteria or (if there is no reference) explain your decision.

On the Figure No 2 line of cut-off is not described.

Line 211-212: Explain, why results of stage 3 are worse than of stage 2. I analyse the Figure No 2 and I observe the same values of percentages related to approval rates for results of stage 2 and stage 3.

Discussion

No comments

Author Response

Dear reviewer, thank you for your suggestions. We try our best to balance your suggestions with those of the reviewers from the previous round. We hope that we have been successful, however, we are available for any clarifications.

The article presented a very important problem related with effectiveness of cleaning and disinfections in emergency units. Studies were good planed and described. I have some comments, especially to the analysis of the results.

Thanks!

Issues:

  1. Abstract

Line23: According to statistically terminology, “p” means a probability value which appears like the result of a specific statistical test. The significance level is defined as α=0.05

-Thanks for that comment. We corrected the error in terminology as suggested by the reviewer.

  1. Line 28: In my opinion, after “assessed” should be semi-colon

-Thanks for that comment. We corrected the error in terminology as suggested by the reviewer.

  1. Introduction

3.1 Line 37: „Healthcare” should be written without space.

-Revised.

Line 40:  There should be used plural form for norovirus.

-Revised.

Line 41: Enterococci is not professional term, please write Enterococcus spp.

-Revised.

Line 41: “spp.” should be written without italics

-Revised.

Line 49: CFU is not a method. If authors want write about method, it should be “CFU counting”

-Revised.

Line 60: Authors write “Few studies…”, but there is only one reference to this sentence.

-Revised

  1. Material and Methods

4.1In my opinion Chart No 1 in not necessary to understand number of evaluations at each stage of the study.

-Thanks for the sugestion. We removed Chart 01. It was just to make it easier for the reader to get into the text.

4.2 Line 148: p value means probability value

We adjusted.

4.3 Line 157: I think that authors missed the parenthesis for: (after-before)

-Revised

4.4 Lines 123-127: This part should be transferred to previously part of this section (after 82 line). There must be added part about interpretation criteria for CFU counting and ATP measuring.

-We transfer and explain the details about the criteria for CFU counting and ATP measuring.

  1. Results

5.1 I am worried about the statistical analysis results. For example, in the table No 1 is presented p-value 0.014 for ATP measurement for women’s toilet flush handle: the median value is 472 RLU, but the distance between minimum and maximum values is high (293 – 1846). It is related probably with acceptance of outlier values. The outlier values are presented also on the figures, for example: on the Figure No 1 is marked value 1552 for door handle in first stage – in my opinion, this value should be removed from the analysis. Please check all analysis results and explain, why did you accept outliers.

-The median is a position statistic and it indicates that 50% of the observed data is below it and the other 50% is above it. In this case, 50% of the data are between 293 and 472 RLU and the remaining 50% are between 472 and 1846 RLU. As the number of data in this distribution is low, the gap between the median and the maximum and / or minimum indicates only the position of the data in relation to the median value. The idea of ​​greater distance between the minimum / maximum values ​​of the median and the discrepancies between the values ​​of average and median indicate greater dispersion of the data, reiterating the application of a non-parametric test for comparative analysis. It is absolutely normal to have these discrepancies, especially in a small N data distribution. In addition, for this reason, we accept all values ​​of the analysis, as we believe that all values ​​should be analyzed according to the method applied, and this was standardized for all data collected. When the method is standardized and the N is small, there is no reason to remove data from the analysis and, in fact, the presence of the outliers is considered important information for the study, since it shows that the ATP quantification is given by a method that causes high variation in the data and is considered to be inaccurate, but with considerable accuracy. The presence of outliers does not affect the result of the statistical analysis, as what is being analyzed as test statistic is the median, a statistic that is not influenced by the outliers.

5.2 I do not understand why authors decided that 250 RLU and 2.5 CFU/cm2 are cut-off values for surface approval. Please, add the reference for this criteria or (if there is no reference) explain your decision.

- Based on several studies widely available in the literature, an acceptable cut-off value (approved) was adopted for the counting of colony forming units (CFU) less than 2.5ufc ‡ / cm2, and for the measurement of the quantity of Relative Light Units (URL) by the ATP method, acceptable values (approved) less than 250 URL. (BOYCE et al., 2011; HUANG et al., 2015; MALIK; COOPER; GRIFFITH, 2003; MULVEY et al., 2011;). LEWIS, et al., 2008

  1. 3 On the Figure No 2 line of cut-off is not described.

-We add the comment: The cutoff values for the ATP count less than 250 RLU and for the UFC count less than 2.5 UFC / cm2

5.4 Line 211-212: Explain, why results of stage 3 are worse than of stage 2. I analyse the Figure No 2 and I observe the same values of percentages related to approval rates for results of stage 2 and stage 3.

-The results of stage 3 were worse than stage 2, because in the long term, that is, after 2 months of the educational intervention, the team did not maintain a good adherence to the educational intervention performed, considering the p values for the UFC counting method . We discussed this result in the discussion section.

Reviewer 2 Report

Thank you for asking me to review this paper. I am not sure why there are inserts in red throughout the paper; has this paper already been reviewed and revised?

The abstract is 290 words; does this comply with Journal word count? Most journals request a maximum of 250 words for the abstract or summary. The abstract itself should contain tangible results with statistical significance and confidence limits.

The English requires some attention throughout the paper.

Stage 1 is possibly better termed ‘baseline’, rather than ‘diagnostic’.

Why were only four hand touch sites chosen and monitored? Are these sites specifically listed in departmental standard operating procedures for cleaning? Does the department have written guidance for cleaners? There seems to have been a choice of cleaning products before the study standardised practice, which suggests some confusion over routine practices.

There is no methodology for microbiology sampling; was this performed with swabs, in which case how were counts quantified? What were the culture and incubation conditions employed by the laboratory? Was it just aerobic colonies examined? Who interpreted the microbiological results?

There were eight hygiene professionals and 16 nurse technicians with cleaning responsibility, but not all undertook the first educational initiative (line 96); was this the case for the educational intervention following stage 1 (line 111)? Who took part in this?

Did staff responsible for cleaning witness the inspection and sampling that took place initially, and during, the study? If they did, they would have known which specific study sites were being monitored.

It is the case that microbiology results from sampling would be affected by the cleaning liquid used for study sites. Quaternary ammonium compounds are not universally effective against all microbes, particularly Gram negative organisms such as pseudomonas. Indeed, ATP signals can also be affected by some cleaning liquids and type of cleaning cloth.

Table 2 does not really provide much information; I would suggest that these data are incorporated into the text.

In conclusion, it is possible that there might have been too few study surfaces chosen to monitor. More data would have provided more robust confirmation on the overall effect on microbiological contamination, while highlighting the reduction in ATP.

Author Response

Dear reviewer, thank you for your suggestions. We try our best to balance your suggestions with those of the reviewers from the previous round. We hope that we have been successful, however, we are available for any clarifications.

  1. Thank you for asking me to review this paper. I am not sure why there are inserts in red throughout the paper; has this paper already been reviewed and revised?

-Yes. This is a revised version based on the contributions of 2 other previous reviewers.

  1. The abstract is 290 words; does this comply with Journal word count? Most journals request a maximum of 250 words for the abstract or summary. The abstract itself should contain tangible results with statistical significance and confidence limits.

-Thanks for that comment. We decrease the size of the summary, and place the p value in the appropriate places.

  1. The English requires some attention throughout the paper.

-Thanks. We request the review by a native professional.

  1. Stage 1 is possibly better termed ‘baseline’, rather than ‘diagnostic’.

-The term has been modified in the relevant places.

  1. Why were only four hand touch sites chosen and monitored? Are these sites specifically listed in departmental standard operating procedures for cleaning?

-For the choice of surfaces, those with the highest frequency of contact were considered as other studies in the literature reported (7,8 9, 10 and 11). We would like to increase the number of surfaces studied, but due to financial limitations this was not possible. This has now been included in the study's limitations.

5.2 Does the department have written guidance for cleaners? There seems to have been a choice of cleaning products before the study standardised practice, which suggests some confusion over routine practices.

- Surfaces are mentioned in general terms in the institution's protocol (critical, semi-critical and non-critical areas). The institution's cleaning protocol had guidance on how to dilute and apply the cleaning products, but it was not clearly and in writing standardized on which surface each product should be used, as a result of this in the study through the intervention educational, the quaternary ammonium based product was standardized for cleaning these 4 evaluated surfaces.

  1. There is no methodology for microbiology sampling; was this performed with swabs, in which case how were counts quantified? What were the culture and incubation conditions employed by the laboratory? Was it just aerobic colonies examined? Who interpreted the microbiological results?

-The methodology for microbiological sampling followed several other studies, according to quote 7,8,14 in the excerpt:

“Contact plates or Rodac Plate®, containing trypticase soy agar (24 cm2) Plast company Labor Ind and Com EH Lab Ltda, Rio de Janeiro, RJ were used to evaluate colony CFUs (7,8,14) The contact plates were pressed on the surface to be evaluated for 10 seconds, then it was introduced into an oven at 37 ° C. The readings took place after 48 hours, only by the main researcher of the study, with the aid of an electronic and digital colony counter (Logenr LS6000), allowing the counting of aerobic colonies. (CLOUTMAN-GREEN et al., 2014) ”.

  1. There were eight hygiene professionals and 16 nurse technicians with cleaning responsibility, but not all undertook the first educational initiative (line 96); was this the case for the educational intervention following stage 1 (line 111)? Who took part in this?

-People who did not participate represent 30% of the professionals responsible for cleaning. The educational intervention occurred only in step 2. During the collections of phase 2, all employees received feedback regarding the results, when requested. That is, even those who did not participate had feedback regarding the cleaning process they were carrying out in phase 2

  1. Did staff responsible for cleaning witness the inspection and sampling that took place initially, and during, the study? If they did, they would have known which specific study sites were being monitored.

-In stage 1, according to lines 113 and 114, the team was unaware of the collections and locations evaluated, to avoid the Hawthorne effect:

“In Stage 1, we aimed to identify a scenario diagnosis, which was the surface C&D with the three monitoring methods: visual inspection, CFU count, and ATP quantification performed by a study researcher. At this stage, the research participants were not notified about the study in order to avoid the Hawthorne effect, thus only managers were informed (13,15-16)”.

In stage 2, an educational intervention was offered in which the objectives of the study were explained and which surfaces were being monitored.

In stage 3, it was performed exactly like stage 1 without feedback

  1. It is the case that microbiology results from sampling would be affected by the cleaning liquid used for study sites. Quaternary ammonium compounds are not universally effective against all microbes, particularly Gram negative organisms such as pseudomonas. Indeed, ATP signals can also be affected by some cleaning liquids and type of cleaning cloth.

-“It is an intervention program, in which the team was trained and offered feedback on the standardization of practices, from products and inputs used, cleaning frequencies and techniques. We cannot affirm the result occurred only due to the standardization of the product, not least because the institution already offered disinfectant based on alkylldimethyl benzyl ammonium chloride.

  1. Table 2 does not really provide much information; I would suggest that these data are incorporated into the text.

-In the first submission, the referee was asked to remove all descriptive information from the results and leave only the tables. We decided to balance and add a little more information from table 2.

  1. In conclusion, it is possible that there might have been too few study surfaces chosen to monitor. More data would have provided more robust confirmation on the overall effect on microbiological contamination, while highlighting the reduction in ATP.

-The evaluated surfaces were chosen according to the frequency of touch, according to several studies available, references 7,8 and 11. However, this is a limitation of the study considering its cost. It should be noted that in new research other surfaces will be evaluated.

Round 2

Reviewer 2 Report

Thank you for responding to the reviewer's comments.

This manuscript is a resubmission of an earlier submission. The following is a list of the peer review reports and author responses from that submission.

Round 1

Reviewer 1 Report

Dear Authors,

Thank you very much for the opportunity to review your manuscript. The study is an observational before and after study with a third period to account for lasting effects . It is methodical sound and its novelty lies within the setting of the emergency department.

While I think that the manuscript is sufficiently crafted to be published. Certain aspects should be addressed prior to that.

Please correct the Manuscript in its revised form with the help of a native speaker. Some parts are not clear or might have been lost in translation.

Major revisions

Please rework the abstract and include results rather than interpretation and explanations of results

Is there any exact information on the participants for the intervention (profession; number of participants in comparison to the department’s staff)

Please shorten the results section (text) considerably! Many results in the text are reported in a redundant manner to figures and charts.

In order to improve upon interpretation of the data you could try creating a weighed score consisting of the three outcome parameters (not only the two that you have in the variation analysis.)

Please consider to combine figures regarding the three outcome parameters (RLU; CFU and visual approval) in a joint figure

Please consider to combine Table 1 and 2 regarding the three outcome parameters (RLU; CFU and visual approval)

Minor revisions

Line 43 would need a literature reference in my opinion.

Please shorten the introduction (line 45 to 49 are not essential to the text; could be used in the discussion)

Line 56 how are the studies limited? Or do you mean there is a limited amount of studies?

Please explain how you accounted for validity in the observers for the visual analysis

Please comment on which basis did you design the individual parts of the intervention?

Please explain how did you define surfaces to be sampled?

Please expand on the limitations of the study (sensitivity of ATP, inter observer variability).

Please rework lines 252-264 and put them into context of existing literature.

Reviewer 2 Report

The purpose of this study is to demonstrate whether the training and education program of cleaning and disinfection could improve the quality of service at ED in a prospective longitudinal fashion by collecting data at 3 stages (before, after #1 (short follow up), after #2 (longer follow up)).

This study has a number of strengths however, I have some concerns about the interpretation of the results. The positive effect of interventions wasn’t sustained in this study based on Figures 1 and 2: Stage 2 appears to be better than Stage 3. Especially when looking at Figure 2, medians were much higher in Stage 3 than stage 1 or 2. Even when looking at table 1, more statistical differences were noticed in Stage 2 than Stage 3.

How did you define success or improvement after this education/training? The definition of this success or improvement was not well written here. Even though the authors looked at the CFU and ATP values, there is no well-defined definition of overall success or improvement of this training course in this study. For example, if there are discrepancies between the CFU and ATP, then what should be done?  

Could you explain more about training and education (for example, how often? or contents)? Did people still use the 3M microfiber before training and education? If not then this improvement could be related to the standardization of 3M microfiber use instead of training and education. 

Methods: -Information regarding the supplier and location (city and state) about the reagents should be consistent throughout the manuscript. For example, “The 3M company”.